# Genetic markers for knee osteoarthritis presence are not associated with disease progression - data from the IMI-APPROACH cohort

Mieke L. M. Bentvelzen[1]*, Paco M.J. Welsing[1], Philippe Moingeon[2],
Simon C. Mastbergen[1], Margreet Kloppenburg[3], Francisco J. Blanco[4], Ida K. Haugen[5],
Francis Berenbaum[6], Hae-Won Uh[7], Mylène P. Jansen[1]☯, Said el Bouhaddani[7]☯

1 Department of Rheumatology & Clinical Immunology, University Medical Center Utrecht, Utrecht, The Netherlands, 2 University Paris Saclay, UFR Pharmacy, Saclay France, 3 Rheumatology, Leids Universitair Medisch Centrum, Leiden, Zuid-Holland, The Netherlands, 4 Grupo de Investigación de Reumatología (GIR), INIBIC – Complejo Hospitalario Universitario de A Coruña, SERGAS. Centro de Investigación CICA, Departamento de Fisioterapia y Medicina, Universidad de A Coruña, A Coruña, Spain, 5 Center for Treatment of Rheumatic and Musculoskeletal Diseases (REMEDY), Diakonhjemmet Hospital, Oslo, Norway, 6 Department of Rheumatology, Sorbonne University, INSERM, AP-HP Saint-Antoine hospital, Paris, France, 7 Department of Data Science and Biostatistics, div. Julius Centrum, University Medical Center Utrecht, The Netherlands.

☯ Both authors contributed equally.
* m.l.m.bentvelzen@umcutrecht.nl

## Abstract

### Objective

Knee osteoarthritis (OA) is a heterogeneous disease with different endotypes and phenotypes, resulting in patients' varying clinical and structural progression. Several genomic markers have been associated with knee OA *presence*. This study aimed to find new associations of these genetic markers with knee OA *progression* and to investigate the risk of knee OA *progression* using a polygenic risk score (PRS).

### Methods

Data from knee OA patients (n = 297) from the IMI-APPROACH cohort with detailed measurements on disease progression were used. Knee OA progression definitions were based on the decrease in minimum joint space width in mm (minJSW; primary outcome), increase in pain on the Knee injury and Osteoarthritis Outcome Score (KOOS), and presence of radiographic OA (based on the Kellgren-Lawrence score) over 24 months. 30 previously reported single nucleotide polymorphisms (SNPs) associated with *presence* of OA irrespective of affected joints or knee OA specifically were investigated. We performed a SNP based genome-wide association analysis using the disease *progression* definitions. Furthermore, a PRS was created using the 30 *presence* SNPs to predict knee OA *progression*.

which permits unrestricted use, distribution, and reproduction in any medium, provided the original author and source are credited.

**Data availability statement:** Data may be obtained from a third party and are not publicly available. In order to gain and govern access to the central APPROACH databases, tranSMART and XNAT, access has to be approved by the APPROACH Steering Committee. Contact information: URCI@umcutrecht.nl.

**Funding:** The research leading to these results has received support from the Innovative Medicines Initiative Joint Undertaking under Grant Agreement no 115770, resources of which are composed of financial contribution from the European Union's Seventh Framework Programme (FP7/2007–2013) and EFPIA companies' in kind contribution. See www.imi. europa.eu and http://www.approachproject. eu. This manuscript reflects the views of the authors and neither IMI nor the European Union and EFPIA are liable for any use that may be made of the information contained herein. The funders had no role in study design, data collection and analysis, decision to publish, or preparation of the manuscript.

**Competing interests:** I have read the journal's policy and the authors of this manuscript have the following competing interests: Outside the current manuscript, PM is employed by Servier and declares no competing interests in relationship with this manuscript; MK reports grants from IMI-APPROACH, grants from Dutch Arthritis Association, royalties from Wolters Kluwer and Springer Verlag, consulting fees (all paid to institution) from Pfizer, Galapagos, CHDR, Novartis, [Peptinov] and UCB, and MK is a member of the EULAR council; FJB reports payment to the institution from Gedeon Richter Plc., Bristol-Myers Squibb International corporation (BMSIC), Sun Pharma Global FZE, Celgene Corporation, Janssen Cilag International N.V, Janssen Research & Development, Viela Bio, Inc., Astrazeneca AB, UCB BIOSCIENCES GMBH, UCB BIOPHARMA SPRL, AbbVie Deutschland GmbH & Co.KG, Merck KGaA, Amgen, Inc., Novartis Farmacéutica, S.A., Boehringer Ingelheim España, S,A CSL Behring, LLC, Glaxosmithkline Research & Development Limited, Pfizer Inc, Lilly S.A., Corbus Pharmaceuticals Inc., Biohope Scientific Solutions for Human Health S.L., Centrexion Therapeutics Corp., Sanofi, MEIJI FARMA S.A., Kiniksa Pharmaceuticals,

## Results

Existing genetic markers for knee OA *presence* were not found to be associated with knee OA *progression*. The PRS of the SNPs for knee OA presence did also not show significant predictive value for knee OA progression. Unexpectedly, nineteen different variants were associated significantly ($P < 5 \times 10^{-8}$) with minJSW decrease. Ten SNPs are located near protein coding genes *PLCL2*, *CDYL2*, and *NTNG1*, and several SNPs are located in or near long non-coding RNAs (lncRNA).

## Conclusions

The 30 OA risk SNPs individually and combined in a PRS are not associated with progression of knee OA in the IMI-APPROACH cohort. 19 different SNPs were associated with minJSW decrease. We demonstrated how to employ multiple bio-informatics tools to, despite a limited dataset, still prioritise potential biomarkers for associations to knee OA progression.

## Introduction

Knee osteoarthritis (OA) is one of the most prevalent types of OA and affects millions of people worldwide [1]. OA is a chronic joint condition with a highly variable disease course, and it is characterised by progressive joint deterioration. Despite significant efforts, no effective treatments are available that can delay or even halt cartilage degeneration [2]. The heterogeneous and unpredictable disease course makes clinical trials and timely personalised treatment approaches challenging. It is hypothesised that different endotypes exist in OA, which are related to variability in disease progression, affected joints, and symptoms [2,3]. To discover targeted drugs, it is essential to find the right pathobiological mechanisms underlying the disease. Therefore, identifying disease subtypes is a promising way forward in the OA field.

One approach to defining these endotypes is through studying genetic variations. Although the heritability of knee OA presence and progression is estimated at 40%−65% and 48%−70% [4,5,6], genetic loci identified so far account for only 10%—20% [7,8]. Over 300 genetic risk loci for OA or knee OA presence have been established, and some of these encode proteins that are active in pathways with available therapeutics. Consequently, these genetic loci may hold potential for therapeutic interventions [7,9,10]. While there is substantial evidence for an inherited genetic predisposition to OA or knee OA, the evidence for a predisposition to progressive disease within knee OA patients is unclear.

Many genetics studies on OA focus on the risk of OA in general or in specific joints rather than progression of disease in patients [7–10]. However, finding genetic variants associated with the risk of *progression* of OA is highly relevant for defining endotypes/phenotypic subtypes of OA. Some clinical risk factors for developing knee OA are also relevant risk factors for disease progression [11]. However, it is unknown whether genetic risk loci for *presence* of OA or knee OA are also relevant for *progression* of knee OA.

Ltd, Fundación para la Investigación Biomédica Del Hospital Clínico San Carlos, Grünenthal, Galapagos; IKH reports personal fees from Novartis, GSK and Grünenthal; FB reports consulting fees from Grünenthal, GSK, Eli Lilly, Novartis, Pfizer, Servier, honoraria for lectures from Viatris and Pfizer, personal fees from AstraZeneca, Sun Pharma, Nordic Pharma and Nordic Bioscience, and is shareholder of 4Moving Biotech and 4P Pharma. The other authors have declared no conflicts of interest. This does not alter our adherence to PLOS ONE policies on sharing data and materials.

In this study, we used patient data from the IMI-APPROACH (Innovative Medicines Initiative Applied Public-Private Research enabling OsteoArthritis Clinical Headway) cohort. Patients with a clinical diagnosis of knee OA were prospectively followed for 2 years. A wide array of data was acquired, including genotyping data, with the aim to discover and predict progressive knee OA endotypes/phenotypes and to support the identification of suitable patients for future clinical trials of disease-modifying osteoarthritis drugs (DMOADs) or treatments (DMOATs) [12]. To gain insights into the heterogeneous nature of OA, the APPROACH project specifically focused on progression definitions regarding widely used clinical outcomes (based on the Knee injury and Osteoarthritis Outcome Score (KOOS) questionnaire [13]) and structural outcomes (based on radiographic minimum joint space width; minJSW [14]).

Our aim was to apply these progression outcomes to find novel associations of previously established genetic markers for *presence* of OA irrespective of affected joints or knee OA specifically with *progression* of knee OA. We calculated a polygenic risk score (PRS) using these previously established SNPs for OA risk to predict progression of knee OA. Finally, we examined novel significant SNPs found during the SNP association analysis. We created an additional PRS using these SNPs and explored their functional implications. Finding genetic loci, either individually or combined, that are associated with disease progression could provide insights into the pathways or molecular mechanisms involved in knee OA progression and enhance our understanding of knee OA pathogenesis.

## Methods

### Study design and participants

Data from the IMI-APPROACH observational cohort study (registered under ClinicalTrials.gov no.: NCT03883568, recruitment period: 2015–2019) was used [12]. The IMI-APPROACH study prospectively followed 297 patients with clinical and/or structural knee OA. All participants provided written informed consent. Demographic, clinical, and imaging data were collected at baseline, 12 months, and 24 months and patient characteristics are presented in Table 1. At the screening visit, an index knee was selected in every patient based on the clinical American College of Rheumatism classification criteria, using history and physical examination [15]. If both knees met the ACR criteria, the most affected knee was selected as indicated by the patient. Posterior-anterior weight-bearing semiflexed knee radiographs of the index knee were obtained to determine the minJSW using Knee Images Digital Analysis software and to grade the Kellgren Lawrence (KL) score [14,16,17]. The radiographs were acquired according to the protocol described by Buckland-Wright *et al.* at baseline and 24 months [18]. Pain was evaluated using the KOOS questionnaire (on a 0–100 scale, with a higher score indicating worse outcomes) [13]. More study details and baseline characteristics have been described previously [12,19].

### Outcome variables

The outcome of interest is knee OA progression. Change in minJSW is a preferred measure for defining structural progression in OA [14,20]. Therefore, we defined the

**Table 1. Clinical characteristics of the IMI-APPROACH study population.**

| Characteristic | Baseline | M24 |
|---|---|---|
| Age, mean (SD) years | 66.5 (7.1) | |
| Female, n (%) | 230 (77.4) | |
| BMI, mean (SD) kg/m² | 28.2 (5.3) | |
| KOOS pain, mean (SD) | 66.1 (18.8) | 68.12 (20.5) |
| minJSW, mean (SD) | 2.52 (1.3) | 2.41 (1.2) |
| KL score 0, n (%) | 53 (18.0) | 38 (16.3) |
| KL score 1, n (%) | 79 (26.9) | 63 (27.0) |
| KL score 2, n (%) | 64 (21.8) | 48 (20.6) |
| KL score 3, n (%) | 87 (29.6) | 74 (31.8) |
| KL score 4, n (%) | 11 (3.7) | 10 (4.3) |

M24, at month 24; BMI, body mass index; KOOS, Knee Injury and Osteoarthritis Outcome Score; minJSW, minimum Joint Space Width; KL, Kellgren and Lawrence.

reduction in minJSW in mm over 2 years as the primary outcome for knee OA progression in this study. Secondary progression outcomes included the absolute change in pain score and binary outcomes concerning the presence of increasing pain (KOOS increase ≥ 5 or ≥ 10 per year with a minimum score of ≥ 35 or ≥ 40 at 2 years respectively, binary), stable high pain (KOOS ≥ 40 at baseline as well as at 24 months), radiographic OA (KL score ≥ 2 at 2 years), and of minJSW decrease of ≥ 0.3 mm/year. These knee OA progression outcomes are in line with the IMI-APPROACH outcome definitions [3].

### Genetic analysis

**Genotype quality control.** Genomic DNA was collected from baseline peripheral blood samples. Genotyping of 2.380.364 single-nucleotide polymorphisms (SNPs) was performed by Newcastle University using the Illumina Omni2.5Exome-8 array v1.5 kit. Quality thresholds can be summarised as follows: call rate per patient ≥ 95%, call rate per SNP ≥ 95%, minor allele frequency (MAF) ≥ 5% and Hardy-Weinberg equilibrium P-value cut-off level ≤ 1E-6. Patients with a heterozygosity rate of ≥±3 standard deviations (s.d.) from the mean heterozygosity rate were removed (n = 6). A set of 54.848 autosomal linkage disequilibrium (LD)-pruned SNPs with MAF > 40% were used to estimate identity by descent (IBD) using PLINK [21]. Patient relationships more closely related than second cousins (proportion of IBD sharing > 0.03125) were identified, and one from each pair was removed (n = 1). The same set of LD-pruned SNPs was used in the assessment of the patients' ancestry. Principal component analysis of these SNPs against three sample populations from the 1000 Genome project (60 samples with European ancestry (CEU), 60 samples with African ancestry (Yoruba, YRI), 90 samples with East Asian ancestry (Japanese/Chinese, JPT)) were used to exclude samples of non-European ancestry (n = 13, see S1 Fig) [22]. Quality control was performed using PLINK v1.9 [21] and R version 3.5.1 (2018-07-02) [23].

**Validation of genetic markers.** Previous Genome-wide association studies (GWAS) in large cohorts discovered over 100 OA risk SNPs [7,9]. We selected all SNPs from these GWAS studies that were significantly associated ($P < 5 \times 10^{-8}$) with the *presence* of OA irrespective of the joint affected or knee OA and that were also present in the genotyping dataset of the IMI-APPROACH study (from now on referred to as 'OA presence SNPs'). This set of 30 OA presence SNPs was investigated in our cohort for an association with knee OA *progression* (see Table 2).

We performed a genome-wide association analysis using the OA progression outcomes. We included the 30 OA presence SNPs, as well as all other SNPs measured in the study, in the analysis (to include other SNPs that may be in linkage disequilibrium with or in the same haplotype block as one of the 30 OA presence SNPs). The analysis was performed

**Table 2. 30 SNPs previously identified to be associated (in GWAS studies with P<5×10⁻⁸) with the presence of OA irrespective of the joint affected or knee OA.**

| Variant | Chromosome: position | EA/NEA | OR | P-value | Nearest Gene | Reference |
|---|---|---|---|---|---|---|
| rs2820436 | 1:219640680 | A/ C | 1.07 | 2.01e-09 | ZC3H11B | [24] |
| rs2605100 | 1:219644224 | A/ G | 1.07 | 4.49e-15 | RP11-95P13.1 | [7] |
| rs7639618 | 3:15216429 | G/ A | 1.43 | 7.30e-11 | COL6A4P1 | [25] |
| rs11177 | 3:52721305 | A/ G | 1.12 | 1.25e-10 | GNL3 | [26] |
| rs6976 | 3:52728804 | T/ C | 1.12 | 7.24e-11 | GNL3 | [26] |
| rs34811474 | 4:25408838 | G/ A | 1.04 | 2.17e-09 | ANAPC4 | [27] |
| rs13107325 | 4:103188709 | T/ C | 1.10 | 8.29e-19 | SLC39A8 | [7,27] |
| rs2066928 | 5:30843787 | A/ G | 0.96 | 1.20e-08 | RPL19P11 | [7] |
| rs10947262 | 6:32373312 | C/ T | 1.31 | 5.10e-09 | BTNL2 | [28] |
| rs7775228 | 6:32658079 | T/ C | 1.34 | 2.43e-08 | HLA-DQB1 | [28] |
| rs12154055 | 6:44449697 | G/ A | 1.03 | 2.71e-08 | CDC5L | [27] |
| rs10116772 | 9:4290541 | C/ A | 1.03 | 3.71e-08 | GLIS3 | [29] |
| rs10974438 | 9:4291928 | A/ C | 1.04 | 7.39e-11 | GLIS3 | [7,27] |
| rs72979233 | 11:74355523 | A/ G | 0.92 | 2.52e-09 | POLD3 | [7] |
| rs10831475 | 11:95796907 | A/ G | 1.08 | 5.89e-12 | MAML2 | [7] |
| rs10842226 | 12:23959589 | A/ G | 1.04 | 4.68e-10 | SOX5 | [7] |
| rs1060105 | 12:123806219 | C/ T | 1.07 | 1.90e-08 | knO1 | [30] |
| rs11842874 | 13:113694509 | A/ G | 1.17 | 2.07e-08 | MCF2L | [31] |
| rs4775006 | 15:58215727 | A/ C | 1.06 | 8.40e-10 | ALDH1A2 | [27] |
| rs12901071 | 15:67370389 | A/ G | 1.08 | 3.12e-10 | SMAD3 | [32] |
| rs9930333 | 16:53799977 | G/ T | 1.05 | 1.52e-09 | FTO | [27] |
| rs34195470 | 16:69955690 | A/ G | 0.95 | 3.13e-13 | WWP2 | [7,30] |
| rs1126464 | 16:89704365 | G/ C | 1.04 | 1.56e-10 | DPEP1 | [27] |
| rs216175 | 17:2167690 | A/ C | 1.04 | 2.74e-12 | SMG6 | [7] |
| rs2953013 | 17:29496343 | C/ A | 1.05 | 3.07e-10 | NF1 | [27] |
| rs10502437 | 18:20970706 | G/ A | 1.03 | 2.50e-08 | TMEM241 | [27] |
| rs10405617 | 19:10752968 | A/ G | 1.03 | 9.33e-11 | SLC44A2 | [7] |
| rs143384 | 20:34025756 | A/ G | 1.08 | 1.01e-23 | GDF5 | [7,27] |
| rs143383 | 20:34025983 | T/ C | 1.17 | 6.20e-11 | GDF5 | [33,34] |
| rs9981884 | 21:40585633 | A/ G | 0.95 | 7.93e-09 | BRWD1 | [7] |

The physical location is given based on GRCh Build 37. Nearest gene: the nearest gene to the specific SNP is given. Abbreviations: rsID, reference SNP cluster ID; GRCh Build 37, Genome Reference Consortium Human Reference 37; EA, effect allele; NEA, non-effect allele.

using linear regression in PLINK 1.9 [21] while adjusting for sex, age, time since OA diagnosis, and baseline minJSW or KOOS pain, depending on the outcome. Binary progression outcomes were analysed with logistic regression using a similar approach. Associations were considered significant at genome-wide level $P<5×10^{-8}$. The 'qqman' R package was used for generating Manhattan and quantile-quantile (QQ) plots [35].

**Functional exploration.** We further explored the SNPs that were significantly associated with OA progression in our analyses using various bioinformatics data sources and open-source software to evaluate their potential (functional) effects. LocusZoom was used to visualise the regions around the significant SNPs that are in linkage-disequilibrium [36]. The effect of SNPs in or near genes on the gene expression levels was evaluated using the baseline RNA-sequencing data from IMI-APPROACH as well (see S1 File) [19]. Furthermore, the significant SNPs were cross-referenced (using

their rsID) in the GTEx Portal (accession number phs000424.v8.p2 on 03/04/2023) to find expression quantitative trait loci (eQTL) that can show previously established associations of the SNPs with protein abundance (up- or down-regulation). Gene expression of genes near significant SNPs was related to the outcome using linear regression. Lastly, a protein-protein interaction (PPI) network was built using the STRING-db (Search Tool for the Retrieval of Interacting Genes, v11.5, http://string-db.org) database to identify networks of proteins coded by genes near the significant SNPs and near previously established SNPs related to knee OA (Table 2) [7,9].

**Polygenic risk score.** A weighted polygenic risk score (PRS) combines the effect of multiple SNPs with a small effect across the genome by multiplying the dosage of the risk alleles (0, 1 or 2 risk alleles) by the effect size of the SNP (log(odds ratio)) and summing across all SNPs. We calculated a PRS combining the 30 OA *presence* SNPs (Table 2) to determine the association of this PRS with the progression of knee OA [7,9]. The PRS was calculated in PLINK v1.9 [21] using the effect size of the SNPs (as previously reported, see Table 2) multiplied by their presence and averaged over the number of non-missing SNPs in individual patients. Linear or logistic regression was then used to determine the association of the PRS with our knee OA progression outcomes while adjusting for sex, age, time since diagnosis of knee OA (of the index knee) and baseline minJSW or KOOS pain. A P-value < 0.05 was considered statistically significant.

## Results

### Participants and descriptive data

The participants in IMI-APPROACH were mostly women (77,4%) and 66.5±7.1 years old on average (Table 1). The majority of participants (55%) had definite radiographic OA (KL score ≥ 2). The mean minJSW was 2.52±1.3 mm at baseline and 2.41±1.2 mm at 24 months; the mean decrease in minJSW over 2 years was 0.17±0.69 mm. Four patients did not agree to genotyping. After data preprocessing, 1.263.798 SNPs and 273 samples (62 males, 211 females) were available for analysis.

### Validation of genetic markers

None of the OA *presence* SNPs were found to be associated with any of the *progression* outcomes (highlighted in green in Fig 1A and S2A, S2C, S2E, S2G, and S2I Fig). However, 19 new variants showed an association with minJSW decrease and reached the significance threshold ($P < 5 \times 10^{-8}$), summarised in Table 3 and shown in Fig 1A and 1B. These 19 SNPs are further explored and associated with minJSW decrease in the next section.

### Functional exploration

While we did not find an association between the 30 OA *presence* SNPs and knee OA *progression*, 19 SNPs were associated with minJSW decrease. Three SNPs on chromosome 3 are in strong linkage disequilibrium: $r^2 > 0.8$ (S3A Fig). LD describes the extent to which one SNP is inherited with or associated with another SNP within the population. These variants most likely represent one locus associated with knee OA progression. The SNPs are located near the *PLCL2* gene, which is involved in cell signalling pathways [37]. On chromosome 16, six SNPs were also in strong LD and located near *CDYL2* (S3B Fig). *CDYL2* is involved in transcription regulation [37]. Several other significant SNPs were in or near *NTNG1*, LINC02619, LINC02379, and *ARLNC1* and were not in LD with other significant SNPs (Table 3).

We examined the effects of the SNPs near *PLCL2* and *CDYL2* on the expression of these genes, but the SNPs were not significantly associated with expression levels (S4 Fig). In contrast, six SNPs near *CDYL2* were found in the GTEx eQTL database to be significantly associated with lower gene expression in muscle skeletal tissues (S5 Fig). Furthermore, *CDYL2* gene expression was significantly lower in patients who had a larger minJSW decrease (S6 Fig, p-value = 0.044). The expression of *PLCL2* was not significantly associated with the minJSW decrease. *NTNG1* gene expression data was not present in the IMI-APPROACH cohort or GTEx eQTL database.

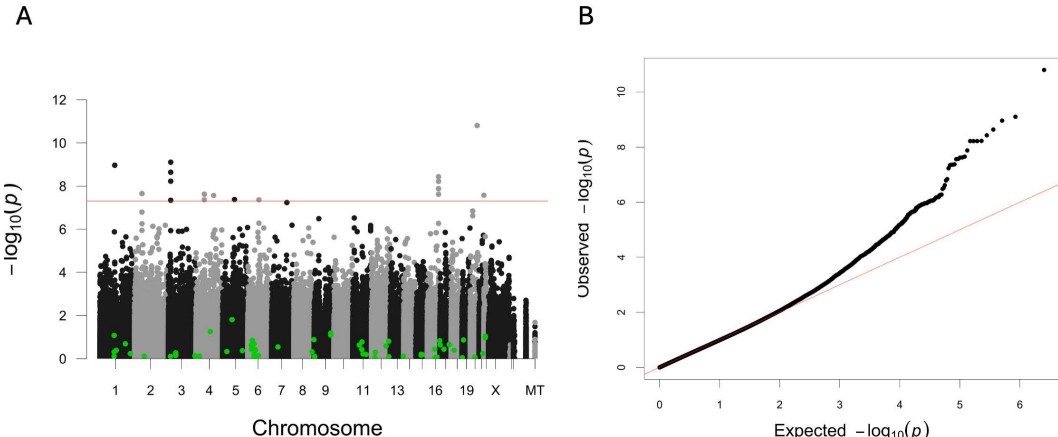

**Fig 1. Validation of genetic markers for risk of OA or knee OA for decrease in minJSW. (A)** Manhattan plot of minJSW decrease at 24 months. The Manhattan plots show the -log10(P) values of all ~1,5 million SNPs against their position. The red line represents the significance threshold ($P < 5 \times 10^{-8}$). Green dots indicate SNPs that have been previously shown in GWAS to be associated with knee OA or OA in general [7,9]. **(B)** Q-Q plot of minJSW decrease at 24 months GWAS. The Q-Q plot shows the genome-wide –log10(P) values of the association analysis. The red line corresponds to the null hypothesis.

**Table 3. Nineteen SNPs with significant association ($P < 5 \times 10^{-8}$) with risk on minJSW decrease in knee OA.**

| rsID | Chr:pos | EA/NEA | Nearest Gene | $\beta$ | SE | P-value | Consequence |
|---|---|---|---|---|---|---|---|
| rs12143353 | 1:107792708 | C/ A | *NTNG1* | −0.56 | 0.09 | 1.08e-09 | Intron Variant |
| rs72806786 | 2:53658706 | G/ A | Na | −0.61 | 0.11 | 2.21e-08 | Na |
| rs6442653 | 3:16924928 | T/ C | *PLCL2* | −0.57 | 0.10 | 4.55e-08 | 2KB Upstream Variant |
| rs6788010 | 3:16929109 | G/ A | *PLCL2* | −0.48 | 0.08 | 5.93e-09 | Intron Variant |
| rs6777965 | 3:16929593 | T/ C | *PLCL2* | −0.51 | 0.08 | 2.27e-09 | Intron Variant |
| rs73146904 | 3:16932989 | A/ G | *PLCL2* | −0.53 | 0.08 | 7.83e-10 | Intron Variant |
| rs76660267 | 4:59800873 | A/ G | LINC02619 | −0.55 | 0.10 | 4.30e-08 | Intron Variant |
| rs116117775 | 4:59983779 | C/ A | Na | −0.54 | 0.09 | 2.35e-08 | Na |
| rs71610153 | 4:126989632 | T/ C | LINC02379 | −0.57 | 0.10 | 2.72e-08 | Intron Variant |
| rs16902074 | 5:85439065 | G/ A | Na | −0.56 | 0.10 | 4.19e-08 | Na |
| rs76865837 | 6:79983511 | A/ T | Na | −0.59 | 0.10 | 4.31e-08 | Na |
| rs77953406 | 16:80833873 | C/ T | *CDYL2* | −0.55 | 0.09 | 1.31e-08 | Intron Variant |
| rs79402702 | 16:80840299 | A/ G | *CDYL2* | −0.53 | 0.09 | 2.38e-08 | 2KB Upstream Variant |
| rs2549732 | 16:80861676 | T/ C | Na | −0.56 | 0.09 | 3.69e-09 | Na |
| rs2549726 | 16:80864621 | C/ T | ARLNC1 | −0.55 | 0.09 | 6.00e-09 | Non Coding Transcript Variant |
| rs9933385 | 16:80864711 | T/ C | ARLNC1 | −0.55 | 0.09 | 6.00e-09 | Intron Variant |
| rs75620223 | 16:80865081 | C/ T | ARLNC1 | −0.55 | 0.09 | 6.00e-09 | Intron Variant |
| rs6021011 | 20:49774249 | T/ C | Na | −0.78 | 0.11 | 1.56e-11 | Na |
| rs73153427 | 22:17718699 | G/ T | Na | −0.37 | 0.07 | 2.66e-08 | Na |

The physical location of the SNP is reported based on GRCh Build 37. The nearest protein coding gene to the SNP (within 2kb upstream of the 5' end of a gene transcript or within 2kb downstream of the 3' end of a transcript) is named. The functional consequence of the SNP on the nearest protein coding gene is reported. Abbreviations: rsID, reference SNP cluster ID; Chr:pos: chromosome: position (base-pairs coordinate); EA, effect allele; NEA, non-effect allele; $\beta$: beta (regression coefficient); SE: standard error of beta estimate; GRCh Build 37, Genome Reference Consortium Human Reference 37.

A protein-protein interaction network did not predict any direct interactions between the proteins near the 19 significant SNPs: *NTNG1*, *CDYL2*, and *PLCL2* (S7 Fig). Similarly, the protein-protein interaction network of the 3 newly identified genes and the genes near the previously identified OA presence SNPs (Table 2) did not show direct interactions (S8 Fig).

**Polygenic risk score**

The PRS based on the 30 OA presence SNPs had a mean ± standard deviation of 0.016 ± 0.006 (normally distributed), indicating a low average genetic predisposition for OA among individuals. The PRS was not significantly related to min-JSW decrease or KOOS pain increase in linear regression analyses (for change in minJSW in mm: β = −1.403, 95% confidence intervals (CI) [−17.084, 14.278], p = 0.862, and for change in KOOS: β = −6.939, 95% CI [−371.357, 357.479], p = 0.970,). Similarly, no significant relations were found for the binary knee OA progression outcomes (S1 Table).

For the sake of completeness, we performed an exploratory analysis using the 19 SNPs newly found to be associated with the minJSW decrease to create a second PRS (−0.040 ± 0.053, mean ± sd). Based on a linear regression analysis, corrected for age, sex, disease duration, and baseline minJSW, a significant association of this PRS with the minJSW decrease was found (β = 7.61, 95% CI [6.07, 9.15], p = 6.899e-19, adjusted R2 = 0.26).

**Discussion**

Contrary to expectations, previous findings on SNPs associated with knee OA presence were not found to be related to the progression of knee OA. Given that the link between a phenotype and single SNPs is generally weak, we calculated a PRS, combining the effects of several OA presence SNPs. This PRS also showed no relation with the progression of knee OA. We hypothesise that different outcomes, i.e., OA presence or OA progression, and their structural or pain characteristics may have partly or even fully different genetic backgrounds.

An unexpected finding was that 19 other SNPs were significantly associated with minJSW decrease over time (Table 3). In reviewing the literature, no previous reports demonstrated an association of these SNPs with OA or knee OA. To interpret these candidate SNP findings, we further investigated them by incorporating various data sources. The GTEx eQTL database revealed that one locus near *CDYL2*, containing six SNPs, showed lower expression of *CDYL2* in muscle-skeletal tissue. *CDYL2* is involved in regulating genome stability, mitosis, and transcription repression [37,38]. Given its role in transcription repression, *CDYL2* could indirectly influence muscle physiology by modulating gene expression of genes involved in muscle function or development. Moreover, skeletal muscle dysfunction has been implicated as a contributing factor to progression of OA [39]. Additionally, we observed a slightly lower expression of *CDYL2* in patients with more decrease in minJSW. This further implies that *CDYL2* might play a role in structural knee OA progression, although likely a small role. However, given the lack of studies linking *CDYL2* to muscle function or OA progression, this hypothesis remains speculative.

We did not observe a significant association between the expression of *PLCL2* and minJSW decrease. However, it was previously found that *PLCL2* is downregulated in knee OA patients with damaged cartilage compared to those with undamaged cartilage [40]. *PLCL2* should be further investigated to determine its role in the progression of knee OA.

Several identified SNPs are located in or near long non-coding RNAs (lncRNA). Although we were unable to investigate the implications of these SNPs, previous studies have shown regulatory effects of lncRNA on pathological processes of OA [41,42]. Further research into lncRNAs could help improve our understanding of molecular mechanisms responsible for knee OA progression.

Known risk factors for OA, such as age, obesity, and sex, are a source of heterogeneity that complicate genetic research in OA. Genetic factors might even affect other prognostic factors (effect modification), further complicating analysis [3]. Our study population included patients at different stages of OA, varying from early to more advanced OA (KL grade 4). While we adjusted for several confounding factors, additional stratification of the population could be beneficial. However, this was not considered feasible due to the relatively small sample size.

One limitation of our analysis is related to the outcome measures. Differences in minJSW are only detectable with certainty if they are > 0.7 mm [20], and the overall changes observed in the cohort were small (the mean decrease in minJSW was −0.16 mm). In addition, the KOOS pain score is a patient-reported outcome measure (PROM). The subjective nature of PROMS, as well as the variability between the patients' perceptions of their symptoms, may have obscured any associations in the analysis.

Most genetic studies on OA focus on the presence of OA rather than looking at subtypes or progression of the disease. Because OA manifests in diverse clinical presentations, often a larger and more diverse study population is needed to find relevant genetic associations, and interpretation of results remains challenging. Future larger studies that focus on more specific OA traits could provide a better understanding of the genetic background underlying OA disease courses.

Although we did not find any associations between genetic factors related to the presence of OA and the progression of knee OA, new associations of SNPs with minJSW decrease were found. To verify that a PRS based on these SNPs would also be significant, an additional PRS was created using these SNPs and indeed showed a significant association with minJSW decrease. These results were calculated in our dataset and not validated externally, so they should be interpreted with caution. We analysed our primary progression outcome on the natural continuous scale (change in minJSW), rather than using a cut-off for presence of progression, to optimally and efficiently analyse the data. One could argue that identifying statistically significant SNPs associated with progression using a strict p-value ($P < 5 \times 10^{-8}$), even with this small sample size, may suggest that there are other SNPs related to progression. Carefully defining OA traits for genetic research may help increase the number of discoveries, statistical power, and clinical interpretability, and ultimately improve future studies and advance our understanding of knee OA. The analysis steps of our study are provided in a GitHub repository (https://github.com/MiekeBvz/IMI-APPROACH_Genomics.git using open-source software) and might be useful for research of genetic associations and further explorations and interpretation of the downstream (functional) effects of genetic markers.

In summary, we found no associations between SNPs related to OA presence and knee OA progression. However, 19 SNPs were newly associated with minJSW decrease and should be further validated to determine their role in knee OA. Given the rising incidence of OA and the lack of effective treatment options, continued efforts are needed to discover the basis of OA and OA progression, enabling patient stratification into more homogeneous subgroups relevant for treatment.

## Supporting information

**S1 Fig. Principal component analysis of the IMI-APPROACH samples against three sample populations from the 1000 Genome project.** A set of 54848 autosomal LD-pruned SNPs with MAF > 40% was used to perform the principal component analysis. Three sample populations from the 1000 Genome project were used to compare to the IMI-APPROACH samples [22]. In black = IMI-APPROACH samples, red = 60 samples with European ancestry (CEU), blue = 60 samples with African ancestry (Yoruba, YRI), and green = 90 samples with East Asian ancestry (Japanese/Chinese, JPT)). (DOCX)

**S2 Fig. Validation of genetic markers for risk of OA for five knee OA progression outcomes.** Manhattan plots showing the -log10(P) values of all ~ 1,5 million SNPs. The x-labels 1–22 represent chromosomes 1–22, 23 is the X chromosome, 24 is the Y chromosome, 25 is the pseud-autosomal region of X, and 26 is mitochondrial chromosomes (0 are unplaced SNPs). The red line represents the genome-wide significance threshold ($P < 5 \times 10^{-8}$). Green dots indicate SNPs that were associated with knee OA or OA in general in previous GWAS (meta-analysis) [7,9]. The Q-Q plot shows the genome-wide –log10(P) values of the association analysis, where the red line corresponds to the null hypothesis. (A) Manhattan plot increasing pain (KOOS score increase of ≥5 or ≥ 10 per year with a minimum score of ≥ 35 or ≥ 40 at 2 years). (B) Q-Q plot increasing pain. (C) Manhattan plot stable high pain (KOOS score of ≥40 over 2 years). (D) Q-Q plot stable high pain. (E) Manhattan plot presence of decrease in minJSW (minimum decrease of ≤ 0.3 mm/year at 2 years).

(F) Q-Q plot presence of decrease in minJSW. (G) Manhattan plot change in pain (change in KOOS score). (H) Q-Q plot change in pain. (I) Manhattan plot presence of radiographic OA (Kellgren Lawrence (KL) ≥ 2 at 2 years). (J) Q-Q plot radiographic OA.

(DOCX)

**S3 Fig. Regional plots of rs73146904 and rs2549732 based on the Genome Reference Consortium Human Build 37.** Association with minJSW decrease (-log10(P)) of SNPs is plotted against the genomic position. The colour represents the pairwise correlation coefficient (Linkage disequilibrium pattern) of each SNP with the most significant SNP (shown as the purple square, in Fig A. rs73146904 and Fig B. rs2549732). (A) The regional plot of rs73146904 shows that several SNPs lay within the near region of *PLCL2*. (B) Regional plot of rs2549732 shows that several SNPs lay within the near region of *CDYL2*.

(DOCX)

**S4 Fig. Comparison of expression levels of PLCL2 and CDYL2 associated with ten significant SNPs.** Comparison of gene expression levels at baseline is shown of PLCL2 and CDYL2 of the patients split into groups based on their genotype. None of the plots show significant up- or down-regulation. The number of patients within a group is noted below the boxplot. The patients with two minor alleles are underrepresented. The black lines indicate the median values. The following SNPs are located near PLCL2 and shown in the plots: (A) rs6442653 versus PLCL2. (B) rs6788010 versus PLCL2. (C) rs6777965 versus PLCL2. (D) rs73146904 versus PLCL2. (E) rs77953406 versus CDYL2. (F) rs79402702 versus CDYL2. (G) rs2549732 versus CDYL2. (H) rs2549726 versus CDYL2. (I) rs9933385 versus CDYL2. (J) rs75620223 versus CDYL2.

(DOCX)

**S5 Fig. Expression quantitative trait loci in muscle-skeletal tissue, associated with six significant SNPs located near CDYL2.** The violin plots of the expression quantitative trait loci (eQTLs) of muscle-skeletal tissue show up- and down-expression of the gene (normalised expression) based on the alleles for six SNPs located near *CDYL2*. The white line in the box plot (shown within the black boxes) represents the median value of the expression. All eQTLs show a significant effect of the genotype on the normalised expression of the gene (all p-values are below the gene p-value threshold of 0.00012).

(DOCX)

**S6 Fig. Scatter-plot of minJSW decrease versus expression of PLCL2 and CDYL2.** The minJSW decrease is in mm over 2 years. Each dot represents a patient. (A) Change in minJSW in mm at 2 years versus the normalised expression of PLCL2 at baseline. The x-coefficient (representing PLCL2) is not significant (P > 0.05). (B) Change in minJSW in mm at 2 years versus the normalised expression of CDYL2 at baseline. The x-coefficient (representing CDYL2) is 0.01 and is significant: P = 0.044.

(DOCX)

**S7 Fig. Protein-protein interaction network between the protein-coding genes near significant SNPs.** Protein-protein interaction network, created with STRING-db, between the protein-coding genes associated with the identified significant SNPs (in red). No known direct associations are shown.

(DOCX)

**S8 Fig. Protein-protein interaction network of proteins near known and newly identified SNPS.** The network, created with STRING-db, includes both protein-coding genes associated with the 19 newly identified significant SNPs (in red), as well as the protein-coding genes as reported along the SNPs identified to be associated with OA in previous GWAS. No direct links are found between genes related to the newly found SNPs and genes near previously established SNPs.

(DOCX)

**S1 Table. Polygenic risk score and OA progression association.** The PRS is calculated using 30 OA presence SNPs. The associations of the PRS with OA progression outcomes, as was found with linear and logistic regression, are summarised in this table. The reduction in minJSW in mm over 2 years and change in KOOS score were used in linear regression. The presence of increasing pain, stable high pain, radiographic OA, and minJSW decrease of ≥ 0.3 mm/year were evaluated using logistic regression.
(DOCX)

**S1 File. RNA-sequencing data.**
(DOCX)

## Author contributions

**Conceptualization:** Mieke LM Bentvelzen, Paco MJ Welsing, Philippe Moingeon, Simon C Mastbergen, Margreet Kloppenburg, Francisco J Blanco, Ida K Haugen, Francis Berenbaum, Hae-Won Uh, Mylène P Jansen, Said el Bouhaddani.

**Data curation:** Mieke LM Bentvelzen, Paco MJ Welsing, Philippe Moingeon, Simon C Mastbergen, Margreet Kloppenburg, Francisco J Blanco, Ida K Haugen, Francis Berenbaum, Hae-Won Uh, Mylène P Jansen, Said el Bouhaddani.

**Formal analysis:** Mieke LM Bentvelzen, Paco MJ Welsing, Hae-Won Uh, Mylène P Jansen, Said el Bouhaddani.

**Funding acquisition:** Paco MJ Welsing, Philippe Moingeon, Simon C Mastbergen, Margreet Kloppenburg, Francisco J Blanco, Ida K Haugen, Francis Berenbaum, Mylène P Jansen.

**Investigation:** Mieke LM Bentvelzen, Paco MJ Welsing, Hae-Won Uh, Mylène P Jansen, Said el Bouhaddani.

**Methodology:** Mieke LM Bentvelzen, Paco MJ Welsing, Hae-Won Uh, Mylène P Jansen, Said el Bouhaddani.

**Project administration:** Paco MJ Welsing, Simon C Mastbergen, Mylène P Jansen, Said el Bouhaddani.

**Resources:** Paco MJ Welsing, Philippe Moingeon, Simon C Mastbergen, Margreet Kloppenburg, Francisco J Blanco, Ida K Haugen, Francis Berenbaum, Hae-Won Uh, Mylène P Jansen, Said el Bouhaddani.

**Software:** Hae-Won Uh, Said el Bouhaddani.

**Supervision:** Paco MJ Welsing, Hae-Won Uh, Mylène P Jansen, Said el Bouhaddani.

**Validation:** Mieke LM Bentvelzen, Paco MJ Welsing, Hae-Won Uh, Mylène P Jansen, Said el Bouhaddani.

**Visualization:** Mieke LM Bentvelzen, Paco MJ Welsing, Hae-Won Uh, Mylène P Jansen, Said el Bouhaddani.

**Writing – original draft:** Mieke LM Bentvelzen, Paco MJ Welsing, Hae-Won Uh, Mylène P Jansen, Said el Bouhaddani.

**Writing – review & editing:** Paco MJ Welsing, Philippe Moingeon, Simon C Mastbergen, Margreet Kloppenburg, Francisco J Blanco, Ida K Haugen, Francis Berenbaum, Hae-Won Uh, Mylène P Jansen, Said el Bouhaddani.

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
