## [Decision Letter · Decision Letter 0]

Dear Dr. Bentvelzen,

We look forward to receiving your revised manuscript.

Kind regards,

Germain Honvo, Ph.D.

Academic Editor

PLOS ONE

Journal Requirements:

“The research leading to these results has received support from the Innovative Medicines Initiative Joint Undertaking under Grant Agreement no 115770, resources of which are composed of financial contribution from the European Union’s Seventh Framework Programme (FP7/2007–2013) and EFPIA companies’ in kind contribution. See www.imi.europa.eu and http://www.approachproject.eu. This manuscript reflects the views of the authors and neither IMI nor the European Union and EFPIA are liable for any use that may be made of the information contained herein.”

“The research leading to these results has received support from the Innovative Medicines Initiative Joint Undertaking under Grant Agreement no 115770, resources of which are composed of financial contribution from the European Union’s Seventh Framework Programme (FP7/2007–2013) and EFPIA companies’ in kind contribution. See www.imi.europa.eu and http://www.approachproject.eu.”

“The research leading to these results has received support from the Innovative Medicines Initiative Joint Undertaking under Grant Agreement no 115770, resources of which are composed of financial contribution from the European Union’s Seventh Framework Programme (FP7/2007–2013) and EFPIA companies’ in kind contribution. See www.imi.europa.eu and http://www.approachproject.eu. This manuscript reflects the views of the authors and neither IMI nor the European Union and EFPIA are liable for any use that may be made of the information contained herein.”

“I have read the journal's policy and the authors of this manuscript have the following competing interests: Outside the current manuscript, PM is employed by Servier and declares no competing interests in relationship with this manuscript; MK reports grants from IMI-APPROACH, grants from Dutch Arthritis Association, royalties from Wolters Kluwer and Springer Verlag, consulting fees (all paid to institution) from Pfizer, Galapagos, CHDR, Novartis and UCB, and MK is a member of the EULAR council; FJB reports payment to the institution from Gedeon Richter Plc., Bristol-Myers Squibb International corporation (BMSIC), Sun Pharma Global FZE, Celgene Corporation, Janssen Cilag International N.V, Janssen Research & Development, Viela Bio, Inc., Astrazeneca AB, UCB BIOSCIENCES GMBH, UCB BIOPHARMA SPRL, AbbVie Deutschland GmbH & Co.KG, Merck KGaA, Amgen, Inc., Novartis Farmacéutica, S.A., Boehringer Ingelheim España, S.A, CSL Behring, LLC, Glaxosmithkline Research & Development Limited, Pfizer Inc, Lilly S.A., Corbus Pharmaceuticals Inc., Biohope Scientific Solutions for Human Health S.L., Centrexion Therapeutics Corp., Sanofi, MEIJI FARMA S.A., Kiniksa Pharmaceuticals, Ltd, Fundación para la Investigación Biomédica Del Hospital Clínico San Carlos, Grünenthal, Galapagos; IKH reports personal fees from Novartis, GSK and Grünenthal; FB reports consulting fees from Grünenthal, GSK, Eli Lilly, Novartis, Pfizer, Servier, honoraria for lectures from Viatris and Pfizer, personal fees from AstraZeneca, Sun Pharma, Nordic Pharma and Nordic Bioscience, and is shareholder of 4Moving Biotech and 4P Pharma. The other authors have declared no conflicts of interest.”

5. In the online submission form, you indicated that your data is available only on request from a third party. Please note that your Data Availability Statement is currently missing the contact details for the third party, such as an email address or a link to where data requests can be made. Please update your statement with the missing information.

Additional Editor Comments:

Please consider thoroughly addressing all the reviewers comments. Please report the revised version of this manuscript strictly following the STROBE Checklist for cohort studies (https://www.strobe-statement.org/). The methods section should particularly include all specific subsections suggested in the STROBE Checklist. Thank you.

Reviewers' comments:

Reviewer's Responses to Questions

**Comments to the Author**

1. Is the manuscript technically sound, and do the data support the conclusions?

Reviewer #1: Partly

Reviewer #2: Partly

Reviewer #3: No

Reviewer #4: Yes

Reviewer #5: Partly

2. Has the statistical analysis been performed appropriately and rigorously?

Reviewer #1: Yes

Reviewer #2: No

Reviewer #3: No

Reviewer #4: Yes

Reviewer #5: Yes

3. Have the authors made all data underlying the findings in their manuscript fully available?

Reviewer #1: Yes

Reviewer #2: Yes

Reviewer #3: Yes

Reviewer #4: No

Reviewer #5: No

4. Is the manuscript presented in an intelligible fashion and written in standard English?

Reviewer #1: Yes

Reviewer #2: Yes

Reviewer #3: No

Reviewer #4: Yes

Reviewer #5: Yes

Reviewer #1: Title: Genetic markers for knee osteoarthritis presence are not associated with disease　progression - data from the IMI-APPROACH cohort

This study was designed to validate genetic markers involved in the disease progression of knee OA, an important disease. Unfortunately, the authors did not obtain the expected results, and the data showing that newly identified SNPs are associated with knee OA are not strong. As the authors comment, knee OA is a very heterogeneous disease, and studies using a larger number of subjects than the relatively small number in this study may be necessary to identify certain trends. On the other hand, the longitudinal evaluation in this study was conducted over a period of 24 months, and the disease progression of knee OA was used as an outcome, which we believe has some value in this regard. However, the definition of "disease progression of knee OA" and past findings related to the obtained results are not sufficiently presented. In addition, the current conclusions seem to contain a lot of content that is not directly derived from the obtained results. I do not believe that it is appropriate to publish the report, at least in its current form.

Major concerns:

1. Conclusions

I believe that only conclusions derived from the results should be stated simply. Especially in the final paragraph of the main text, considerations and assertions are conspicuous. I suppose the message to the reader will become clearer and more appropriate by moving the assertions to the other part of the discussion section.

2. Inclusion criteria for target patients

Patients with OA according to the ACR criteria were included, but with widely varying disease status, ranging from early OA with no clear objective findings to advanced OA (KL grade 4). Although there may be limitations due to the small number of patients included in the study, we suggest that an analysis limited to patients with KL grade 2 or higher (or grade 1 or higher) should be considered.

3. Definition of disease progression in knee OA

"Minimum joint space width" and "KOOS pain" have been used. Are these commonly used in past papers? I believe that these definitions are central to this study and should be stated in such a way that it is clear that the authors' definitions are appropriate. In particular, regarding "minimum joint space width," it should be clearly stated in the text how lateral joint space/medical joint space/PF joint space, etc. were treated. Although references are provided, I believe that it would be appropriate to include in the text the method used to take the images and the measurement sites used for the measurements.

4. Discussion

The results of this study are simple but important, and we believe that they should be thoroughly reviewed in light of previous findings. Are there no reports prior to this study regarding the two major results of "SNPs related to knee OA progression" and "Relationship and comparison between presence and progression of knee OA (including genetic and other factors) "? Please discuss the results of this study after citing the key reports, as I believe there are some, at least with respect to the latter.

Specific points:

Background information on the subject patients

Very limited information is provided. Please consider adding information such as past history of knee trauma (post traumatic OA) and KOOS score other than pain.

30 SNPs

How were these extracted? Were there any selection criteria?

Or were they arbitrarily selected by the authors from previous literature?

Limitation

Please add "limitation" to the "discussion section".

Reviewer #2: Thank you for an important study in a field which deserves attention. Although the framework of the study seems to be of a good quality, the manuscript is a little unpolished and confusing. In general, the M&M and results sections could use more structure.

As far as I can tell, a full GWAS on the progression of OA has been performed, with several outcomes available, both radiological change (minJSW) and PROMS data (KOOS). The authors wanted to see if previous knee OA associated SNPs would be replicated, and to see if these 30 SNPs could be used as a PRS for knee OA progression. They have also done several functional analyses based on the results.

Major comments:

1. The manuscript is confusing in its current form. I would present the data as a knee OA progression study with the primary and secondary outcomes from the current dataset in a more structured way, leaving the previous knee OA SNPs as a later, separate paragraph along with the PRS. As of now, it is difficult to understand the difference between the 30 known SNPs and the completely separate 19 SNPs from the current GWAS that was actually significant. I suggest switching Table 2 (previous knee OA SNPs) with Table S1 (Current associated SNPs). I would also use fig S1 A and B (Manhattan and QQ-plot for minJSW change) as main figures in the text. Some of the functional data as well as figures could also be included in the main body of the paper.

2. I would like there to be more details on the minJSW measurements, the SD is quite high compared to the average change between 0 and 24 months. A figure such as a box plot would be nice, as well as some discussion around the limitations of the measurement, precision etc.

3. Technically, as several outcomes have been tested, the threshold for statistical significance should reflect this. With 6 outcomes, it should be (5x10-8)/6 if using Bonferroni correction.

4. As the study is borderline in terms of power with regards to some of the outcomes (as apparent in several QQ-plots), the manuscript could do with elaboration on the general limitations of the study, PROMS data etc.

Minor comments:

1. Some superfluous use of parentheses in the text, like (knee) OA.

2. Write a little bit more about the functional analyses other than "incorporating various data sources" (p 8, line 160-161)

Reviewer #3: In this manuscript, the authors tested 30 published SNPs and found no association with knee osteoarthritis progression. This paper has several major issues, my comments are as below:

1. The authors only investigated 30 previously reported SNPs for OA progression, which is obviously too few and unredeemable. Ideally, the authors should generate their own genome-wide SNPs array data and test it again in IMI-APPROACH cohort.

2. Each section of this paper is too brief and lacks sufficient depth. To enhance the quality and rigor of the paper, more content should be added to each section. For example, the introduction section could be expanded by including a more detailed background on the research topic, providing context to the field.

3. The SNP location was based on GRCh Build 37; however, the more advanced version, GRCh Build 38, has been released in 2013, which was more than ten years ago. The authors should definitely utilize the latest genome build and try to re-run the analysis.

Reviewer #4: In this study, the authors assessed whether a polygenic risk score (PRS) based on 30 genetic variants that have been associated with risk of knee osteoarthritis (OA) could be used to predict progression of knee OA. The authors found no association of the PRS to the progression of knee OA, but detected 19 significant genetic variants associated with a decrease of minimum joint space width over a 24 month period, an indication of worsening OA.

It is not surprising that the PRS is not significantly associated with knee OA progression given that none of the 30 PRS variants reached statistical significance in the GWAS. However, I have concerns about how these 30 SNPs were selected, please see my comments below.

The focus could be shifted away from assessing the polygenic risk score which was designed to predict the presence of OA, rather than the progression of OA, although this can still be included in the study as it is still valuable knowledge that variants for risk of knee OA do not overlap with those associated with progression.

Overall, there is enough data here for this study to be an independent examination of genetic markers (independent of the 30 known risk variants) associated with kneeOA progression. It could be further improved with more thorough examination of the 19 significant SNPs, and depending on data availability, the addition of transcriptomic analyses of changes over time or within-population differences that contribute to worsening OA. With some major revisions, I would recommend this study for publication.

Please see my specific comments below:

Abstract:

- KOOS not defined

- Please italicise gene names on line 37 and throughout the manuscript.

Methods:

- Was BMI recorded at 24 months? Is the change in BMI or high baseline BMI associated with worsened OA progression?

- References in table 2 are not in reference list and does not follow same citation style. Please state if P values and OR from Table 2 are from the present study or from the cited source. It is also unclear as to how these 30 OA presence SNPs were selected. Presuming that Boer et al., 2021 (reference 6) is currently the largest GWAS study for knee OA, why were their 31 significant kneeOA SNPs not used for assessment and PRS in the current study? Additionally SNPs rs2605100, rs10831475 and rs9981884 were listed under the TJR (total knee and/or hip replacement) phenotype, while rs13107325, rs216175 and rs10405617 were listed under AllOA phenotype instead of kneeOA in Boer et al., 2021.

- There were 297 patients in the cohort (Line 70), however blood was only collected from 288 patients (Line 90), why where 9 samples excluded?

- How was relatedness assessed? Please provide kinship values and cutoffs used.

- Further details are needed for the ancestry assessment. How many reference samples were used to compare against the cohort, and were they derived from publicly available data? If so, please cite the source. How many variants were used for the ancestry assessment? It would be good to show a PCA plot of the samples in the supplement, including those from the reference ancestries.

- Was multiple testing correction done for the regression analyses?

Results:

- Considering that none of the 30 OA presence SNPs were associated with progression outcomes, it is not surprising that the PRS was not significantly associated with progression either given it is a small set of SNPs. It would be interesting for the authors to examine a PRS based on the 19 significant SNPs from the minJSW decrease GWAS. Is it possible to predict whether a patient will have decreased minJSW based on these 19 SNPs?

- How many base pairs away were considered a ‘nearest gene’ for Table S1? There are several genes in the table with “Na” for nearest gene, how far away is the closest gene? Please state the odds ratio with 95% confidence intervals or beta coefficient with standard error for each SNP as well. It may be worth examining LD around all the top hits and investigating what other genes are nearby. P < 5 × 10−8 is quite a strong association considering the sample size so the other associations shouldn’t be discounted.

- As supplementary S1 and S2 forms a major component of this study, these sections belong in the main text. Regarding the available RNA-sequencing data, when were these samples collected (baseline, 12 months, 24 months)? If there is access to RNA-seq data for all three time points, of even for two time points, it would be valuable to assess the transcriptomic changes with worsening knee OA. If there is only one sampling for RNA-seq, it would be valuable to compare transcriptional differences within the population, for patients with the greatest progression compared to those with the least.

- Please state an r^2 value in which “strong” LD is defined, or report the range of r^2 values for the SNPs that are in strong LD.

Discussion:

- Citation for line 163 lacking (“..which is implicated in the regulation of genome stability as well as mitosis…”)

- I could not access https://github.com/MiekeBvz/IMI-APPROACH_Genomics.git

- Any discussion points for PLCL2? What is the function of this gene?

- As above, the other significant hits should not be ignored, I think some discussion is warranted around them, including those near or in lncRNAs since ncRNAs have shown a role in progression of OA (e.g. Ji et al., 2023, Am J Hum Genet 110(4):606-624, PMID: 36868238).

Supplementary:

- Figure S1 – not sure why there is chromosome 0 in the Manhattan plots – are they possibly standards used by Illumina? Also please rename chromosomes 23-26 as X, Y, and MT or note it in the figure legend. The axes should also be adjusted for figures C-L, the y-axis limit could be set to 10. In the figure legends, please state for each GWAS if it was a linear or logistic regression.

- Figure S5 – please state the units for change in minJSW.

Reviewer #5: In this study, Bentvelzen et al. asked if previously identified genetic markers of knee Osteoarthritis (OA) presence are also risk makers of knee OA progression. Authors accessed a cohort of 297 patients from the IMI-APPROACH prospective clinical study, and used the study progression outcomes to test for genome-wide associations with markers present in the Illumina Omni2.5Exome-8 array v1.5 kit. The study details and clinical characteristics are remitted to references 2 and 9 in the manuscript (Deveza et al, 2019; van Helvoort et al, 2021)*. The primary progression outcome was decrease in minimum joint space width (minJSW) in mm over 24 months. The researchers genotyped 288 of the patients, and 273 of these samples (62 male, 211 female, all of European ancestry) along with 1.263.798 SNPs with minor allele frequency > 5% passed GWAS quality control parameters and were included in the analysis. The authors identified, within the genomic array kit, 30 variants as previously associated with OA presence and investigated for their association with OA progression outcomes in their cohort. They used linear regression for the continuous OA progression outcomes and logistic regression for binary progression outcomes with the PLINK v1.9 software. Calculations were controlled for “sex, age, time since OA diagnosis, and baseline minJSW or KOOS pain depending on the outcome”. The authors have also calculated a weighted polygenic risk score (PRS) on the 30 selected variants and tested for its association with the same outcomes using the same software and controlling for the same covariates. The results showed no evidence of association between the variants (which had been previously associated with OA presence) with the disease progression, and likewise, the PRS of these variants was not associated with progression. The lack of association suggested that different outcomes (i.e., OA presence vs. OA progression) could have different genetic backgrounds. Indeed, additional analysis revealed that 19 other SNPs, which had not been previously associated with knee OA presence, were associated with the knee OA progression outcome: minJSW decrease. Upon bioinformatic mining of the associated SNPs and interrogation of another dataset from the same cohort - IMI-APPROACH cohort gene expression - the authors selected two genes (genes PLCL2 and CDYL2) each located in the vicinity of clusters of associated variants, that were in linkage disequilibrium, for further exploration. There was no association between the OA progression outcomes with gene expression (non-normalized counts) for PLCL2, but a small trend was observed for the gene CDYL2. Interestingly, upon exploration of the GTEx eQTL database, 6 of the variants located around gene CDYL2 significantly associated with OA progression were also associated with lower expression of the gene in the muscle skeletal tissue.

The study warrants publication, not only due to the significance of reporting “negative results”, but also because it underscores, as the authors suggest, the value of exploring a broad range of disease traits and (endo)phenotypes beyond primary disease classification. This approach could lead to uncovering of novel targetable pathways - currently hidden within more heterogeneous patient cohorts - potentially improving patient care. The authors found a secondary association with progression of OA, though they advise caution in interpretation due to the study’s sample size. I believe that with a few concerns addressed, confidence in the study’s findings could be strengthened.

Overall, my first point of criticism is that the analysis lacks sufficient detail in the description of datasets and methods and even on the choices made during the analysis. This lack of transparency limits clarity, interpretability and reproducibility. With a few adjustments and a more thorough discussion, this study could not only gain credibility but also potentially open up exciting new avenues for research. My major points of concern are the following:

1. Power calculations should be explicitly shared. The authors mention that the study is relatively small, but do not provide quantitative data to justify this claim. It is essential to include power calculations in the manuscript for the reader to assess the significance of the reported associations.

2. Sufficient control for confounding effects needs to be better addressed. In the discussion, the authors write: Lines 166-9: “Known risk factors contributing to the likelihood of OA, such as age, obesity, and sex (…). While we adjusted for these factors, further stratifying the population could be beneficial, but it was not feasible in our relatively small sample.” – however, in the methods section, authors don’t report to have adjusted for obesity: Lines 102-3: “(…) while adjusting for sex, age, time since OA diagnosis, and baseline minJSW or KOOS pain depending on the outcome.” This lapse could be a minor issue, but a closer look at the results got me intrigued on whether the associations reported could be confounded with traits which have been previously reported as associated with the genes highlighted as associated and which could likely be risk factors for OA progression. Namely, SNPs in the vicinity of the genes CDYL2 and ARLNC1 have been previously associated with traits such as “heel bone mineral density”, “waist-hip ratio”, “BMI-adjusted waist-hip ratio” and “body height” amongst others (as seen on a quick search on the GWAS catalog for both genes ARLNC1 and CDYL2, at https://www.ebi.ac.uk/gwas/genes/ARLNC1 and https://www.ebi.ac.uk/gwas/genes/CDYL2 ). Some of these variants could be in linkage disequilibrium with the variants associated with OA progression and even if not, they could still be confounded. The authors should describe which relevant clinical variables they have access to, rather than referencing other publications. They should adjust for all likely confounders present in the data, particularly those that are known or likely risk factors for OA and have previously been associated with the genes identified in this study. Additionally, the authors should account for the fact that some covariates may not be independent. Given concerns about power loss, they could prioritize confounders based on effect size, and if not adjusting for all confounders, at least include those with the largest effect sizes.

3. A more in-depth exploration of the gene PLCL2 and its previously associated traits may provide a basis for a functional hypothesis or reveal additional confounding effects. Notably, markers within or near the gene PLCL2 have been extensively linked with immune traits, including (auto-)inflammatory conditions such as Rheumatoid Arthritis (RA) and Psoriasis (https://www.ebi.ac.uk/gwas/genes/PLCL2.) It is also known that conditions such as Psoriatic Arthritis (PsA) are frequently undiagnosed and can be misdiagnosed for RA and OA (Saalfeld et al, 2021, https://www.ncbi.nlm.nih.gov/pmc/articles/PMC8572231). This warrants the inclusion of inflammatory markers (e.g., serologic) and disease history (e.g., psoriasis and other inflammatory diseases) as covariates in the GWAS analysis. After such adjustments, whether the association is confirmed or excluded as confounded by inflammatory disease covariates, the rigor of the results will be improved, and perhaps the authors will be more confident in advancing the study to a more comprehensive stage. For example, identifying an associated variant could facilitate earlier diagnosis of PsA in a patient of this study, allowing for earlier treatment and potentially reducing permanent joint damage. Alternatively, it could inform future studies aimed at developing diagnostic tools and impacting patient care.

4. The gene expression data must be normalized. Without adjusting for read depth, comparison across different libraries is not valid, rendering both the small association with CDYL2 expression and the lack of association with PLCL2 expression unreliable. Additionally, I could not find any explicit details in the manuscript regarding the type of gene expression experiment used (it appears to be RNA-Seq) and it is not mentioned in the methods section the kind of biological material used. These issues should be addressed and clarified.

Other minor opportunities for improvement, include:

1. I commend the authors to share the analysis steps of the study and analysis in a GitHub repository (Line 180). However, the GitHub repository either does not exist or is currently not publicly accessible. This issue should be addressed.

2. I recommend putting figures pertaining to results directly answering the main questions of the study in a more prominent position instead of supplementary data. I would also suggest including all the methods description in a single methods section.

3. As already stated, a) data parameters relevant to the study should be revealed; b) all methods for which there are results shown should be adequately and explicitly described.

4. General image readability should be improved (e.g., Figure S2 is not readable even after zooming in.)

5. Images and plots should be adequately labeled (e.g., in figure S3, the gene names could be annotated as title instead of only on the description of the figure. Overall visualization could be improved, maybe with violin plots instead of boxplots. Indeed, a good example of how to show this data is exemplified in Figure S4.)

6. A better description of the figures and tables should be attempted (e.g., in Table S1, explain what “Na” means – maybe by explaining the criteria used for “nearest gene”?)

7. Authors should search for minor grammatical inconsistencies across the manuscript.

* Perhaps the authors would like to confirm if both these references are correct.

**Do you want your identity to be public for this peer review?** For information about this choice, including consent withdrawal, please see our Privacy Policy

Reviewer #1: No

Reviewer #2: **Yes: ** Kaya Kvarme Jacobsen

Reviewer #3: No

Reviewer #4: **Yes: ** Pamela Xing Yi Soh

Reviewer #5: **Yes: ** Clara F. Alves-Pereira

---

## [Author Response · Author response to Decision Letter 1]

25 Nov 2024

The response to the reviewers is upload as an attachment, because it exceeds 20,000 characters. The attachment is called 'Response to Reviewers'.

---

## [Decision Letter · Decision Letter 1]

Dear Dr. Bentvelzen,

Thank you for submitting your manuscript to PLOS ONE. After careful consideration, we feel that it has merit but does not fully meet PLOS ONE’s publication criteria as it currently stands. Therefore, we invite you to submit a revised version of the manuscript that addresses the points raised during the review process.

We look forward to receiving your revised manuscript.

Kind regards,

Germain Honvo, Ph.D.

Academic Editor

PLOS ONE

Journal Requirements:

Additional Editor Comments :

The Editor reiterates his previous recommendation to report the manuscript strictly following the STROBE Checklist for cohort studies (https://www.strobe-statement.org/). In particular, the methods section should include all specific subsections suggested in the STROBE Checklist. Please include section titles as suggested by STROBE : 1) Study design; 2) Setting; 3) Participants; 4) Variables; etc., with slight adaptations if necessary. Please note that this will improve readability and clarity of the manuscript. Also, please report Funding information in the Acknowledgments Section of your manuscript, not in the Cover letter.

Reviewers' comments:

Reviewer's Responses to Questions

**Comments to the Author**

Reviewer #1: All comments have been addressed

Reviewer #2: All comments have been addressed

Reviewer #3: All comments have been addressed

Reviewer #4: All comments have been addressed

Reviewer #5: All comments have been addressed

2. Is the manuscript technically sound, and do the data support the conclusions?

Reviewer #1: Yes

Reviewer #2: Yes

Reviewer #3: Yes

Reviewer #4: Yes

Reviewer #5: Yes

3. Has the statistical analysis been performed appropriately and rigorously?

Reviewer #1: Yes

Reviewer #2: No

Reviewer #3: Yes

Reviewer #4: Yes

Reviewer #5: Yes

4. Have the authors made all data underlying the findings in their manuscript fully available?

Reviewer #1: Yes

Reviewer #2: Yes

Reviewer #3: Yes

Reviewer #4: Yes

Reviewer #5: No

5. Is the manuscript presented in an intelligible fashion and written in standard English?

Reviewer #1: Yes

Reviewer #2: Yes

Reviewer #3: Yes

Reviewer #4: Yes

Reviewer #5: Yes

Reviewer #1: All comments have been appropriately addressed, and I judge that it is acceptable to consider publication in this academic journal.

Reviewer #2: I thank the authors for their thorough review of this paper. It is now much more coherent and easier to read and follow along in the process. They have also addressed the concerns of myself and the other reviewers. However, in the revision process, they have added some PRS data that I would like to address as a major concern:

The topic of PRS and their usage and utility in clinical work is much debated. Still, a basic principle of PRS is to use a baseline dataset(s) to determine risk variants and their effect sizes and applying this to an independent target dataset(s) to see if the risk is transferable and quantifiable. In this paper, after not finding a knee OA presence PRS to be relevant to knee OA progression, the authors have used the 19 significant SNPs in their progression dataset as a PRS within their own dataset. This leads to an overfitting of the model, as seen with the exceptional p-value. Also, in pure scientific terms this approach makes little sense. Taking known risk variants for OA progression in a sample and testing if they are predictors of OA progression in the same sample will of course give positive results. Thus, I feel that this part of the paper should be removed. This concerns page 11-12 Line 209-212 and page 13 line 249-251.

Minor comments:

1) A few points in the abstract: Page 2 line 24-25: "Some genomic markers were previously associated with knee OA presence." Please rephrase along the line of " Several genomic markers have been previously associated with knee OA presence.". The current sentence is a little unpolished.

In the abstract and also later in the text I would advice to write something like "Kellgren Lawrence (KL) OA score" or in some other way specifying what the KL score is scoring. Also in the abstract line 31-32: "We associated SNPs in a genome-wide association analysis..." is not very well formulated. A more correct phrase would be "We did a SNP based genome-wide association analysis.." or "We tested for association..". Finally, in line 36, "Unexpectedly, nineteen new variants" is meant to separate the presence SNPs from the progression SNPs. In order to not confuse these variants with de novo etc, I would advice to say "nineteen different" or "separate" or something along those lines.

2) In the main text, page 6 line 124, you also use the term "were associated with" when describing the testing itself. As above, please rephrase.

3) Page 11 line 188-189, please take the text in the parenthesis out as a separate sentence.

4) Page 11 line 195-197: You state that CDYL2 is associated with lower gene expression in skeletal muscle tissues. Although very speculative, could the role of this gene in knee OA progression be related to muscle strength or muscle control? Typically knee OA patients often have effect on symptoms by doing physical therapy increasing their muscle strength and control around the knee?

5) Page 12 line 219: Please remove parenthesis.

Reviewer #3: The authors have addressed my comments and improved the manuscript now. I suggest the paper to be accepted at its current form.

Reviewer #4: The authors have made significant improvements to the paper and have addressed all my main concerns. I have a few minor comments on the current version:

- Italicise gene names on line 37 in abstract

- For lines 46-67, I would suggest rephrasing to:

”Despite significant efforts, no effective <disease-modifying> treatments are available that can delay or even halt cartilage degeneration [2].”

- Line 73: add “We calculated a polygenic risk score <(PRS)> using these previously established <single > for OA … “

- Line 109: define “s.d.”

- Regarding using the IBD cut-off of 0.03125, why was this cutoff selected? This seems quite stringent to me, please justify. How did you choose which one of a related pair to remove? Was this based on genotype missingness?

With the ancestry assessment, what was used to determine “non-European ancestry” (Line 116)? On the PCA plot (Fig S1) I can see two samples grouping with East Asian ancestry, and 11 samples that are admixed European-Asian, which could confound your GWAS. Please justify, and change the axes labels on Fig S1 to PC1 and PC2.

- Line 118, remove “in”: Previous Genome-wide association studies (GWAS) in large cohorts discovered <in> over 100 OA risk SNPs [7,9]

- Lines 192-193: Italicise gene names

- Fig S5: Is it possible change this to GRCh37 assembly base positions to be consistent with the rest of the paper? Labelling with rsIDs would help with cross referencing with Table 3.

- Table S1: p-value “0,970” - please change to decimal.

- The authors could consider depositing their GWAS results in the GWAS catalog (https://www.ebi.ac.uk/gwas/), to assist future research on the same topic.</in></single></disease-modifying>

Reviewer #5: (No Response)

**Do you want your identity to be public for this peer review?** For information about this choice, including consent withdrawal, please see our Privacy Policy

Reviewer #1: No

Reviewer #2: **Yes: ** Kaya Kvarme Jacobsen

Reviewer #3: No

Reviewer #4: **Yes: ** Pamela Xing Yi Soh

Reviewer #5: **Yes: ** Clara Alves-Pereira

---

## [Author Response · Author response to Decision Letter 2]

17 Jan 2025

A rebuttal letter that responds to each point raised by the academic editor and reviewers is uploaded as separate file labeled 'Response to Reviewers'.

---

## [Decision Letter · Decision Letter 2]

Dear Dr. Bentvelzen,

We look forward to receiving your revised manuscript.

Kind regards,

Germain Honvo, Ph.D.

Academic Editor

PLOS ONE

Journal Requirements:

Reviewers' comments:

Reviewer's Responses to Questions

**Comments to the Author**

Reviewer #2: All comments have been addressed

2. Is the manuscript technically sound, and do the data support the conclusions?

Reviewer #2: Yes

3. Has the statistical analysis been performed appropriately and rigorously?

Reviewer #2: No

4. Have the authors made all data underlying the findings in their manuscript fully available?

Reviewer #2: Yes

5. Is the manuscript presented in an intelligible fashion and written in standard English?

Reviewer #2: Yes

Reviewer #2: Thank you for revising the manuscript and addressing all of my comments. However, I still disagree with your choice to include the PRS calculations on the origin sample set.

**Do you want your identity to be public for this peer review?** For information about this choice, including consent withdrawal, please see our Privacy Policy

Reviewer #2: **Yes: ** Kaya Kvarme Jacobsen

---

## [Author Response · Author response to Decision Letter 3]

25 Apr 2025

See 'Response to Reviewers' document.

---

## [Editor Report · Decision Letter 3]

Genetic markers for knee osteoarthritis presence are not associated with disease progression -  data from the IMI-APPROACH cohort

PONE-D-24-06311R3

Dear Dr. Bentvelzen,

We’re pleased to inform you that your manuscript has been judged scientifically suitable for publication and will be formally accepted for publication once it meets all outstanding technical requirements.

Kind regards,

Germain Honvo, Ph.D.

Academic Editor

PLOS ONE

**Additional Editor Comments** :

Dear Author,

Thank you for submitting your revised manuscript.

I would like to draw your attention to the fact that, in providing comments and suggestions on your manuscript, the sole objective of both the reviewers and the Editor is to help you improve it.

Therefore, your role is to try to revise your manuscript according to reviewers’ comments or if necessary, to explain why you disagree with any specific comments. Please note that your role is not to tell the Editor whether he/she should accept or not your manuscript.

Please kindly keep this in mind for future submissions.

Best regards,

Germain Honvo,

Academic Editor.
---

## [Editor Report · Acceptance letter]

PONE-D-24-06311R3

PLOS ONE

Dear Dr. Bentvelzen,

I'm pleased to inform you that your manuscript has been deemed suitable for publication in PLOS ONE. Congratulations! Your manuscript is now being handed over to our production team.

Kind regards,

on behalf of

Dr. Germain Honvo

Academic Editor

PLOS ONE